# Food Costs of Children and Adolescents Consuming Vegetarian, Vegan or Omnivore Diets: Results of the Cross-Sectional VeChi Youth Study

**DOI:** 10.3390/nu14194010

**Published:** 2022-09-27

**Authors:** Eva Hohoff, Helena Zahn, Stine Weder, Morwenna Fischer, Alfred Längler, Andreas Michalsen, Markus Keller, Ute Alexy

**Affiliations:** 1Department of Nutritional Epidemiology, Institute of Nutritional and Food Science, University of Bonn, 44225 Dortmund, Germany; 2Research Institute of Plant-Based Nutrition, 35444 Gießen/Biebertal, Germany; 3Faculty of Human Resources, Health & Social Work, University of Applied Sciences (FHM), 33602 Bielefeld, Germany; 4Faculty of Health, Gemeinschaftskrankenhaus Herdecke, Witten Herdecke University, 58313 Herdecke, Germany; 5Institute of Social Medicine, Epidemiology and Health Economics, Charité-Universitätsmedizin Berlin, 10117 Berlin, Germany

**Keywords:** food costs, children, adolescents, vegetarian diet, vegan diet, omnivore diet

## Abstract

The aim was to analyse the total food costs and the impact of food groups on total food costs among vegetarian, vegan and omnivore children and adolescents in Germany. Based on three-day weighed dietary records of 6–18-year-old children and adolescents of the VeChi Youth Study, the total daily food costs and food group costs (both EUR/day, EUR/1000 kcal) of a vegetarian (n = 145 records), vegan (n = 110) and omnivore (n = 135) diet were calculated. Minimum retail prices of 1000 empirically selected foods reported in the dietary records were linked to individual food intakes. The group differences were analysed using ANCOVA or Kruskal-Wallis tests. Vegans had the highest energy adjusted total food costs at 2.98 EUR/1000 kcal, vegetarians the lowest at 2.52 EUR/1000 kcal. Omnivores also had significantly higher costs than vegetarians with 2.83 EUR/1000 kcal/1000 kcal (*p* = 0.01), but the total costs did not differ significantly between omnivores and vegans (EUR/d and EUR/1000 kcal). Compared to vegetarians, vegans had significantly higher expenditures (EUR/day) on fruit (*p* = 0.0003), vegetables (*p* = 0.006), dairy alternatives (*p* = 0.0003) and legumes/nuts/seeds (*p* = 0.0003). Expenditure on starchy foods was significantly higher in the vegetarian or vegan than in the omnivore diet (*p* = 0.0003). Omnivores spent a quarter of their total food costs on animal source foods (25%), which is equivalent to the sum of food costs for legumes/nuts/seeds, dairy alternatives and meat alternatives in vegans and additionally dairy in vegetarians. The VeChi Youth Study indicated that financial constraints are not necessarily a barrier to switching to a more plant-based diet.

## 1. Introduction

Food-based dietary guidelines in Europe mostly refer to health aspects, while sustainability often plays a minor role [1,2]. Besides ecological aspects such as the emission of greenhouse gases or land use, sustainability also has an economic and social dimension. Thus, a healthy and ecologically sustainable diet must be affordable for individuals of different socio-economic status [3], which is especially important for families with children. To promote both health and environmental sustainability, plant-based diets are recommended, including omnivore, vegetarian or vegan food patterns [4]. However, little is known about the food costs of plant-based diets compared to an omnivore diet, reflecting the economic sustainability. In a mixed diet, animal foods make up a large portion of the food cost. The estimation of the food costs of an omnivore diet for German children and adolescents in 2009 showed daily costs of about 2 EUR/1000 kcal, based on minimum food prices. In this analysis, meat (13% of total daily costs), milk and dairy products (12%) and beverages (13%) had the highest share [5]. In a recent modelling study on the global and regional costs of dietary patterns, the relative affordability was largest for vegetarian and vegan diets that focused on legumes and whole grains [3]. In contrast, another modelling study from New Zealand came to different results. Here, the vegan dietary pattern yielded the highest food costs compared to the more inexpensive current diet [6]. In an older study, individual food costs calculated on the basis of the real-life data of a vegetarian and omnivore diet in German adult women resulted in the lowest food costs in a dietary pattern with a high proportion of plant-based and unprocessed foods [7,8]. However, a vegan diet excluding all animal source foods was not considered in this study. Furthermore, these results might be not comparable to other population groups, such as children, and the data are outdated as food prices change over time due to inflation, changes in production costs, increased global competition or changes in subsidies [9,10]. For example, the prices of food and non-alcoholic beverages in Germany increased by about 10% between 2015 and 2020 [9]. The largest price increase related to the food groups fish (15%), meat (14%) and fruit (14%), followed by vegetables (11%) and dairy products (11%). The smallest price increase was observed for confectionery (2%) [9].

In addition, secular trends in dietary habits can affect daily food costs regardless of the dietary pattern. In recent years, not only has a trend towards plant-based diets been observed [11,12], but also meat and dairy intake was declining in predominantly omnivore populations [13,14,15] and being replaced by plant foods. At the same time, plant-based dairy and meat alternatives are becoming increasingly popular [16,17,18]. In direct comparison, the prices of these alternative products are considerably more expensive than the original animal-derived foods [19].

Thus, higher food costs are supposed to be a barrier for switching to a more plant-based diet, in particular for socially disadvantaged groups, as food prices are one key factor for food choices [20,21]. This might explain the observed higher prevalence of a vegetarian or vegan diet among population subgroups with a high socio-economic status [22,23].

Hence, the aim of this study was to calculate and compare food costs among children and adolescents consuming a vegetarian, vegan or omnivore diet based on real-life data. Hereto, data from the VeChi Youth Study, a cross-sectional study conducted in Germany from October 2017 to January 2019 [24,25], were used and linked to retail food prices collected in 2021.

## 2. Materials and Methods

### 2.1. The VeChi Youth Study

The primary objective of the VeChi Youth Study was to collect and analyse dietary intake and anthropometric data, as well as to assess the nutrient status using blood and urine biomarkers from vegetarian (excluding meat and fish), vegan (excluding all animal foods) and omnivore children and adolescents aged 6–18 years. Details of the VeChi Youth Study have been described in detail elsewhere [24,25,26].

### 2.2. Study Sample

The VeChi Youth Study included 401 children and adolescents aged 6–18 years who were examined in three study centres in Germany. Of the participants, 390 (boys n = 169, girls n = 221) completed a three-day weighed food record. Accordingly, data from 110 vegans (28%), 145 vegetarians (37%) and 135 omnivores (35%) were evaluated for the present analysis.

### 2.3. Dietary Assessment

Dietary intake was recorded using three-day weighed dietary records as described elsewhere [14]. In short, all of the foods and beverages consumed, as well as the leftovers, were weighed and recorded over three days using electronic kitchen scales. If exact weighing was not possible (e.g., for out-of-home meals), semi-quantitative household recording (e.g., spoons and cups) was allowed. Missing data were obtained by the study staff by requesting the information from parents via email. For commercial food products, e.g., ready-to-eat meals and meat or dairy alternatives, the exact brand name was reported. The energy and nutrient composition of such food products were calculated by recipe simulation based on the nutrient and ingredient declaration [27]. Hence, it was possible to collect not only staple food prices (e.g., apples and milk) but also brand-specific product prices (e.g., vegan meat alternatives or lemonades) in this project.

### 2.4. Food Price Collection

A total of 3046 different foods and food products were reported in the dietary records of the VeChi Youth Study. After excluding condiments (e.g., salt and pepper) and dietary supplements, 2866 staple foods and composite food products remained. To reduce the effort for price collection to a feasible level, a representative sample of 1000 food items was selected, analogous to previous studies [5,28]. Hereto, those 800 food items with the largest mean consumption amounts were selected in a first step (basic foods). Subsequently, from the remaining foods, a random sample of 200 food items was selected. With this approach, we captured foods that have an impact on costs due to the high quantity which is eaten, as well as foods that are eaten only in small quantities. Food prices were collected in February and March 2021. Due to the COVID pandemic restricting retail store visits, the prices were largely determined on the websites of two popular German supermarket chains (www.rewe.de and www.kaufland.de accessed on 12 February 2021) and one discount chain (www.aldi-onlineshop.de accessed on 19 March 2021). If the prices of specific branded products were not available on these shop websites, the prices were collected in other online shops. If more than one kind of similar food was offered in the same point of sale, e.g., for the staple foods of apples or cow milk, the minimum price (EUR/100 g) at the respective point of sale was recorded. To ensure comparability, special offer prices were excluded and prices of the same packaging sizes were collected. For food consumed out of the home, the prices on the website of the respective restaurant chain or a comparable restaurant was collected. For out-of-season varieties of fruit and vegetables (e.g., berries), the price of the respective deep-frozen food was determined.

### 2.5. Calculation of Food Costs

First, the average prices (EUR/100 g) of different points of sale were calculated for each food item, if necessary. All reported food items were assigned to one of 14 food groups (Table 1).

Since only the prices of 1000 selected food items were collected, the missing 1866 prices had to be estimated. For this purpose, the missing food item prices were replaced by the respective average price of the respective food group.

The individual mean of the total daily food costs (EUR/day) was calculated by summing up the multiplied product of food price (EUR/g) and food consumption (g/day, individual mean of three record days). As energy requirements and intake differ by age and sex, the daily food costs were additionally standardised to the individual total daily energy intake (TEI) (EUR/1000 kcal TEI).

In addition, the food group costs were calculated according to the same procedure, both as EUR/day and EUR/1000 kcal of energy intake from the respective food group (EUR/1000 kcal FG), as well as total daily food group cost shares (% of food group costs in total food costs per day).

### 2.6. Assessment of Covariables

In addition to sex (boys/girls), also the age of participants (years), body mass index-standard deviation score (BMI-SDS), socio-economic status (SES, high/medium/low) and physical activity (MET-min = metabolic equivalent task-minutes) were considered as potential confounders. The Winkler index [29] was used to determine SES, combining parental education, parental profession and the total net household income (1–7 points, each) assessed by a questionnaire. In the case of different values of mother and father, the higher value was used. The index is categorised into low (Winkler Index 1–7 points), medium (Winkler Index 8–14 points) and high (Winkler Index 15–21 points) SES. Missing values of covariates (n = 11) were replaced by the respective median of the corresponding diet group. The BMI-SDS was calculated using the LMS method based on the German reference percentiles for children and adolescents [30] using measured values for body weight and height [24]. Physical activity was assessed by a questionnaire based on the validated Adolescent Physical Activity Recall Questionnaire [31] which included questions on organised and non-organised sport activities.

### 2.7. Statistical Analysis

All statistical analyses were carried out with SAS^®^ 9.4. The sample characteristics were described as median and quartiles (Q1; Q3) for continuous variables due to a lack of the normal distribution of most variables. The non-parametric Kruskal-Wallis test was used to examine the differences of continuous characteristics. The categorical variables were presented using absolute (n) and relative frequencies (%).

The differences in categorical variables between diet groups were tested by a Chi^2^ test or Fisher’s exact test. An analysis of covariance (ANCOVA) was performed to assess group differences of the total daily food costs (EUR/day and EUR/1000 kcal TEI) between vegetarian, vegan and omnivore participants. All models were adjusted for sex (male/female), the age of the participants (years), BMI-SDS, SES (high/middle/low) and physical activity (MET minutes). In the case of the total daily food costs (EUR/day), models were additionally adjusted for TEI (kcal/d). An ANCOVA was also performed to assess group differences of food group costs among those food groups with a low number of non-consumers (vegetables, fruits, starchy foods, fats/oils, sweet and snack foods as well as beverages, each calculated as EUR/day and EUR/1000 kcal FG). If necessary, some zero values were replaced by the smallest value > 0 (winsorised). For dairy intake, an ANCOVA was performed to assess the group differences between vegetarians and omnivores only.

Due to the high number of non-consumers of legumes/nuts/seeds, dairy alternatives and meat alternatives, convenience foods and eggs, the differences between the vegetarian, vegan and omnivore participants were analysed using the non-parametric Kruskal-Wallis test, both for EUR/day and EUR/1000 kcal FG. Pairwise comparisons were performed using the non-parametric Wilcoxon-Mann-Whitney *t*-test. No statistical tests were performed for meat/fish, as this food group was consumed only by one diet group.

Using the false discovery rate, the proportion of false positive dependencies (significant *p*-values) due to multiple testing was calculated and corrected with the Benjamini–Hochberg procedure [32]. The significance level was set at a *p*-value < 0.05

## 3. Results

### 3.1. Sample Characteristics

The present analysis includes all the available complete dietary records of 390 study participants (boys n = 169; 43.3%) (Table 2). The age of the study participants did not differ between groups. Most of the participants (n = 278, 71.3%) were from families with a high SES. The SES differed significantly between the diet groups (Fisher’s Exact Test, *p* = 0.0058). Children and adolescents with a vegan diet came more often from families with a medium SES and less often from families with a high SES than children and adolescents with a vegetarian or omnivore diet. Both the TEI and BMI-SDS were highest in the omnivore group and lowest in the vegan group (*p* < 0.05). With regard to physical activity, the diet groups did not differ.

### 3.2. Total Daily Food Costs

The median total daily food costs were highest for vegan participants (4.79 EUR/day), followed by omnivore (4.75 EUR/day) and vegetarian participants (4.37 EUR/day). Total daily food costs differed significantly between diet groups (*p* = 0.036) (Table 3). Pairwise comparison showed no significant difference of total daily costs between vegan and omnivore participants. These results were confirmed when the total daily food costs were standardised for TEI (EUR/1000 kcal).

### 3.3. Food Group Costs

Food group costs (EUR/day) differed significantly between diet groups (*p* < 0.0045), with the exception of sweet and snack foods, convenience foods and oils/fats (Table 3).

When standardised for energy intake from the respective food group (EUR/1000 kcal FG), diet group differences were statistically significant for legumes/nuts/seeds, dairy alternatives and meat alternatives, as well as for sweet and snack and convenience foods (*p* < 0.015) (Table 3).

Regardless of the diet group, the daily food cost shares were highest for starchy foods and fruits (Figure 1 and Appendix A). Among omnivore participants, meat/fish had the fourth highest cost shares (14% of the total daily food costs, 0.43 EUR/day). The food group costs for vegetables accounted for around 9–12% of daily food costs, with the highest shares among vegan participants and lowest shares among omnivore counterparts.

Per day, vegetable food costs of vegans (0.57 EUR/day) were significantly higher than for vegetarians (0.36 EUR/day; *p* = 0.0006) and omnivores (0.38 EUR/day; *p* = 0.0003). When standardised for energy intake, these differences were no longer significant. In all diet groups, vegetables were the food group with the highest cost per 1000 kcal (>8 EUR/1000 kcal), followed by fruit (>3.5 EUR/1000 kcal).

Omnivore participants spent significantly more on dairy, with 0.42 EUR/day, than vegetarian participants, with 0.26 EUR/day (*p* = 0.015). In contrast, vegan participants spent 0.45 EUR/day on dairy alternatives, and vegetarian participants spent 0.08 EUR/day (*p* = 0.0003). Less than half of the omnivore group consumed dairy alternatives (zero median food costs).

The food costs of meat alternatives were lower among vegetarians (0.17 EUR/day) and vegans (0.25 EUR/day) than the food costs of meat for omnivores (0.57 EUR/day). The energy-related expenses for meat alternatives (>5 EUR/1000 kcal) and dairy alternatives (2.75 EUR/1000 kcal among vegetarians and 4.61 EUR/1000 kcal among vegans) were higher than for the original animal-derived products.

Legumes/nuts/seeds contributed only a little to the daily food costs, but they differed significantly (vegan 6%, vegetarian 3% and omnivore 1%). The daily food costs for sweet and snack foods differed not significantly between diet groups. But there was a statistical significance regarding EUR/1000 kcal between vegetarians and vegans (*p* = 0.0042), with higher expenditures for sweet and snack foods by vegans.

The median food costs for sweet and snack foods ranged from 2.11 EUR/1000 kcal among vegan and 2.41 EUR/1000 kcal among vegetarian and omnivore participants. The highest median beverage costs were found among omnivores (0.57 EUR/day), which were significantly higher than those among vegetarian (0 EUR/1000 kcal; *p* = 0.0155) and vegan participants (0 EUR/1000 kcal; *p* = 0.0022).

The food costs for convenience foods did not differ significantly. When standardised for energy intake from the respective food group, the difference between vegans (4.43 EUR/1000 kcal) and both other groups (vegetarian: 2.97 EUR/1000 kcal; *p* = 0.0391 and omnivore: 1.88 EUR/1000 kcal; *p* = 0.0036) was significant.

## 4. Discussion

### 4.1. Main Results

To the best of our knowledge, this is the first study comparing the food costs of children and adolescents on a vegan, vegetarian and omnivore diet based on reports of self-selected diets and retail prices. Our data showed that the vegetarian diet was the most inexpensive dietary pattern, independent from TEI. In all diet groups, starchy foods, fruit, sweet and snack foods, beverages and vegetables contributed the most to daily costs. For omnivore participants, meat/fish also made a significant contribution. Protein foods, i.e., the sum of legumes/nuts/seeds, dairy alternatives, meat alternatives, dairy (vegetarian and omnivores only) and meat/fish (omnivores only), contributed a quarter of the total food costs, independent from diet group. The share for dairy alternatives of vegans in the total costs corresponded to the share of dairy products for omnivores.

### 4.2. Total Daily Food Costs

According to our study, families spent 2.52 EUR–2.98 EUR/1000 kcal a day on food for each child. The above mentioned evaluation using a similar approach with food prices from 2009 calculated food costs at 1.84 EUR–2.00 EUR/1000 kcal per day for omnivore children and adolescents [5]. Thus, the food costs calculated in this study were considerably higher, independent from the dietary pattern. Similar to our findings, a recent study in Germany which investigated the food costs of four week sample menus using minimum retail prices estimated food costs per day of 5.17 EUR/day (2.59 EUR/1000 kcal) and 5.69 EUR/day (2.85 EUR/1000 kcal) for 10–13 year old girls and boys on a vegetarian diet, respectively. In this study, the food costs of a vegan diet were about 1 EUR/day higher (6.17 EUR/day for girls, 6.97 EUR/day for boys) [33]. This higher estimated total daily food costs may be due to a higher assumed energy requirement (PAL 1.6, i.e., 2000 kcal/day for girls, 2200 kcal for boys). In the VeChi Youth Study, the TEI corresponded to the energy intake reference for a PAL of 1.4 [34]. Energy requirements and thus the amount of food needed is the most important determinant of total food costs [5]. Therefore, daily food costs increase with age and are higher for boys than for girls [5,33]. That is why the present evaluation considered both daily food costs and food costs, standardised to energy intake.

Both total daily food costs and standardised total food costs differed between the diet groups and showed a small but statistically significant benefit of vegetarian diets. Our results thus confirm the above mentioned German study from 2009, in which a vegetarian diet was associated with lower food costs in adult women than a diet that included the consumption of meat and fish [7]. As mentioned in the aforementioned study on the food costs of sample menus, a vegetarian pattern (as well as an omnivore diet consisting of fresh food) was the most inexpensive pattern. A vegan diet was estimated to be more expensive [33].

However, the total daily cost differences in our study between dietary patterns were only small: the median total daily food costs of the omnivore and the vegan diets exceeded the daily food costs of a vegetarian diet by 0.31 EUR/day and 0.46 EUR/day, respectively. Per month, this results in a total difference of 9.30 EUR and 13.80 EUR. Nevertheless, for socially disadvantaged families or families with several children, these differences can be decisive for the choice of diet. However, it should be noted that the difference between the total daily food costs of vegan and omnivore diets was not statistically significant. Hence, a change from an omnivore diet to a more sustainable vegetarian diet could even save money.

Furthermore, the wide interquartile ranges (IQR) of energy standardised food costs in all diet groups are worth mentioning (vegetarian participants: IQR = 1.04 EUR/1000 kcal, vegan participants: 1.01 EUR/1000 kcal and omnivore participants: 0.94 EUR/1000 kcal). Hence, the food costs of the dietary pattern overlap and there is a financial margin to make each diet pattern more inexpensive, for example, by the selection of less expensive foods within a food group. Hence, a shift towards a vegan diet should not cause necessarily any additional costs.

### 4.3. Food Group Costs

The total daily food costs of a diet are partially determined by the amounts of food groups consumed. Because changes in the consumption of food groups, such as a shift from animal source to plant foods, also affect the consumption of other food groups [7], it was not possible to identify single food groups responsible for the observed differences in daily costs.

Starchy foods had the highest budgetary demand in our sample in the vegetarian and vegan group. This reflects the high contribution of this food group to TEI [25]. The high share of fruit and vegetables costs confirm the current state of the literature [3,33], in particular when expressed as costs per calorie [21]. However, this calorie-based approach does not take into account that consumers do not only buy food to meet their energy needs, but also for other reasons, such as individual preferences in taste or health aspects. Sweet and snack foods were also relevant for the energy and food cost shares. Vegans had significantly lower cost shares for this food group (11%) than the participants of the other two diets (14% each). This can be attributed to the lower consumption of sweet and snack foods compared to omnivore and vegetarian participants [25]. In the aforementioned German modelling study [33], on the other hand, the cost shares of the sweet and snack food group were 3% in a vegan diet, but only 1% in a vegetarian diet. However, this result can be attributed to the fictive nature of the sample menus.

In the VeChi Youth Study, dairy was consumed less by participants on a vegetarian diet than on an omnivore diet [25]. That is why vegetarians spent less money on this food group than omnivores. Instead, vegetarians also spent 5% of their daily food costs on dairy alternatives; vegans spent 8%. Both vegetarians and vegans spent less money on meat alternatives than omnivores for meat, although energy standardised costs of meat alternatives are higher than of meat. Meat, fish and eggs as well as dairy account for substantial food cost shares in omnivore diets [3,33,35].

It is recommended for vegetarians and vegans to increase their consumption of soy, legumes, nuts and seeds to provide a sufficient amount of protein, iron and zinc [12,36]. The nutrient profiles of meat alternatives and dairy alternatives are not comparable with the original animal-derived foods, and the nutrient content is variable depending on the ingredients used [37,38,39]. In a direct price comparison (EUR/100 g), they are often more expensive than the original animal source foods [38]. Nevertheless, in the VeChi Youth Study, meat food costs were comparable to the sum of costs for plant protein sources, i.e., legumes/nuts/seeds and meat alternatives in vegans. Willits-Smith et al. [35] even predicted a food cost reduction of 10% when replacing meat with legumes, seeds, nuts and soy foods in a New Zealand modelling study.

Besides the food group intake amount, food choices within a group, for example the type of fruit or vegetable or the degree of processing, affect food costs. That is why it can be argued that cost estimates based on real-life data are more valid than cost estimates based on sample menus or modelling studies. In the present study, this became particularly clear for beverages, as omnivores differed significantly from vegetarians and vegans in terms of daily beverage costs. Beverages even accounted for the largest share of costs next to sweet and snack foods in an omnivore diet. This can be explained by the fact that vegan but also vegetarian study participants consumed more inexpensive beverages such as tap water, while omnivores more often consumed expensive bottled water and soft drinks (data not shown).

Regarding the total costs for processed foods such as convenience foods, meat alternatives, etc., vegans showed the highest values, which also contributed to the higher daily total costs.

Another important aspect of food selection is the distinction between conventional and organic produced foods. Organic food, as well as animal source food assuring special animal welfare, e.g., pasture milk or meat from animals raised in larger stables, are more expensive than conventional food products [33]. About a quarter of the study participants in the VeChi Youth Study who followed a vegan or vegetarian diet reported buying more than 75% organic food, but only about 12% of those were following an omnivore diet [24]. In our survey, we could not distinguish whether unprocessed staple foods, e.g., fruits, vegetables or starchy foods, were from organic or conventional production. However, the price collection of special brand products considered the production procedure. The associated price effects may contribute to the observed differences in food costs.

### 4.4. Strengths and Limitations

The present analysis has some methodological strengths and limitations that need to be discussed. As mentioned before, a cost calculation based on real-life consumption data yields more plausible cost estimates than a calculation based on sample menus. Another important strength is the detailed brand-specific dietary survey, which enabled a very accurate and diet group-oriented price survey. Our price survey, which was conducted in different supermarkets, should provide a representation of food prices as comprehensive as possible at the time of the survey.

A limitation of our method is the short time span of the price survey, which does not capture seasonal effects, e.g., for the price of vegetables and fruit. In addition, the survey of all 2866 food items recorded in the food logs was not feasible within the framework of this project. However, the present approach is a proven method for calculating food costs [5,28]. Moreover, due to the COVID-19 pandemic and the resulting lockdown measures, a price survey was only possible online. A price survey in stationary retail might have led to slightly different total daily food costs. However, this is a random error that affects all diet groups equally. The effect of food price differences on total costs was excluded in this evaluation, as minimum prices were used.

A further limitation is the lack of representativity of the study sample. The high SES of most participating VeChi Youth Study families are consistent with the known sociodemographic characteristics of vegetarians and vegans [11]. Although statistical models were adjusted for a self-reported SES, some residual confounding cannot be excluded and the limited generalisability of the results must be kept in mind when interpreting them.

Last but not least, it should be mentioned that we only assessed direct food costs, but did not consider further cost associated with nutrition, e.g., costs for purchase, storage or preparation.

## 5. Conclusions

Our data showed that a real-life vegetarian diet pattern was the least expensive compared to an omnivore and vegan diet pattern among German children and adolescents. Overall, vegans had lower cost shares for sweet and snack foods and beverages than vegetarians and omnivores. In contrast, vegans had higher cost shares of vegetables, legumes/nuts/seeds, fruits, dairy alternatives and meat alternatives than vegetarians and omnivores. These differences added up to the observed total daily food costs differences. Overall, our results confirm the findings of previously published studies [3,6,7,33,35], although they do not correspond to them in all aspects. Even though the food market is becoming increasingly global, there are differences in food prices between countries [33], e.g., due to subsidies or taxes. The comparability of international studies therefore is limited.

Our data have some policy implications [3,35] as taxes and subsidies influence food prices. For example, in Germany, dairy and meat are taxed at a reduced value-added tax (VAT) rate of 7%, while dairy alternatives and meat alternatives are taxed at 19%. To achieve significant reductions in obesity prevalence and health care costs [40], German health professional societies proposed in 2017 a VAT reform whereby adipogenic foods would become more expensive and fruit and vegetables would become cheaper [41]. This proposal has not yet been implemented. However, such VAT reform should not only consider the health effects of foods, but also promote environmentally sustainable diets.

Overall, our evaluation shows that plant-based diets do not have to be more expensive than an omnivore diet. Financial constraints are not necessarily a barrier to switching to a more plant-based diet.

## Figures and Tables

**Figure 1 nutrients-14-04010-f001:**
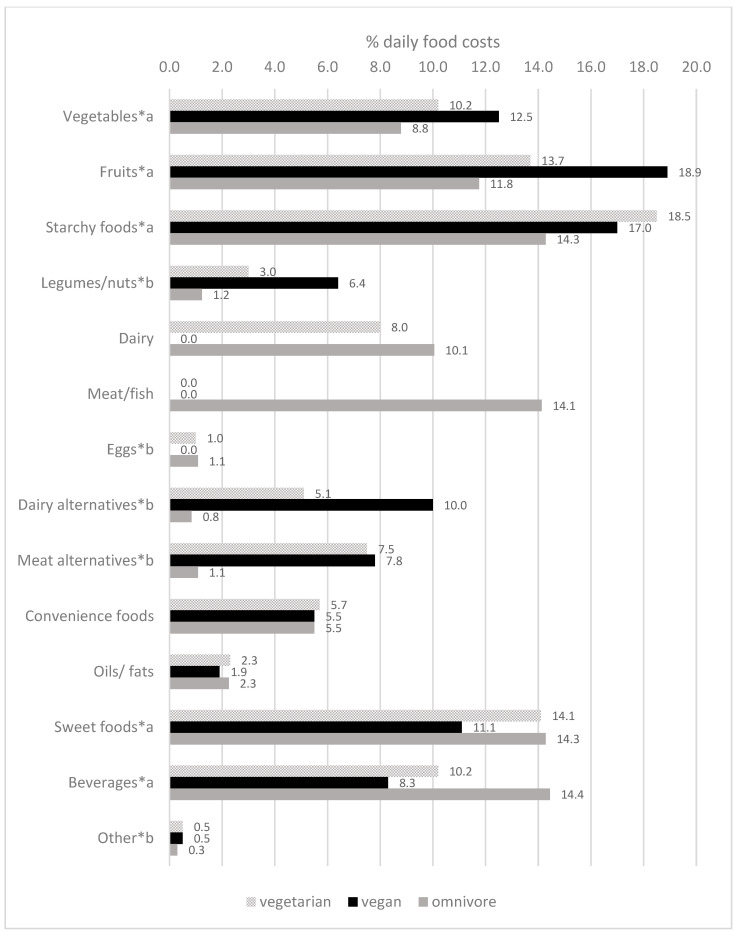
Daily food group costs as % of total daily food costs of participants of the VeChi Youth Study (n = 390) stratified by diet group (* indicates significant differences between diet groups; performed using ^a^ Ancova and ^b^ Kruskal-Wallis test).

**Table 1 nutrients-14-04010-t001:** Classification of food groups (modified according to Alexy et al. [24]).

Food Group	Included Foods
Vegetables	fresh, frozen and dried vegetables, mushrooms, fresh herbs, olives, vegetable juices, ready-made salads, vegetable products, preserves
Fruits	fresh, frozen and dried fruit, juices ^1^, smoothies, squash, preserves
Starchy foods	bread, rolls, flour, doughs, semolina, flakes, breakfast cereals, muesli mixes, rice, pseudo cereals, pasta, dumplings, potatoes, french fries, croquettes, potato dumplings, mashed (powdered) potatoes
Legumes/nuts/seeds	peas, beans, lentils, lupins, soybeans, also as flours, falafel, nuts (also nut butter, nut puree) and seeds (e.g. sesame, sesame puree), roasted almonds
Dairy	milk, cream, cheese, quark (products), fresh milk products, milk-based drinks, milk-based desserts
Meat/fish	meat, sausage, ham, meat products, fish, fish products, seafood
Eggs	hen’s egg, scrambled egg, fried egg
Dairy alternatives	plant-based alternatives for milk, yoghurt, quark and cheese, silken tofu
Meat alternatives	meat and sausage imitates, roast meat, tofu, soy cutlets
Convenience Foods	frozen pizza, canned soups, ready-made sauces, food from snack bars, vegetable spreads based on pulses, vegetables, nuts or avocado
Oils/fats	oils, butter, margarine, lard
Sweet and snack foods	sugar, syrups, sweet breads, jams, nut nougat cream, biscuits, nibbles, sweet foods, chocolate, ice cream, etc.
Beverages	water, coffee, tea, alcoholic drinks, soft drinks
Others	water for cooking, vinegar, mustard

^1^ Commercial spritzers were divided into juice and water and assigned to the respective groups.

**Table 2 nutrients-14-04010-t002:** Sample characteristics of VeChi Youth Study participants (n = 390) stratified by diet group.

	Total	Vegetarian	Vegan	Omnivore	*p*-Value ^1^
Participants	390	145 (37.2)	110 (28.2)	135 (34.6)	
Girls	221 (56.7)	87 (60.0)	73 (66.4)	61 (45.2)	0.0023
Age (in years)	12.5 (9.2; 16.3)	12.4 (9.2; 16.0)	12.8 (9.0; 16.9)	12.3 (9.5; 16.2)	0.8741
Energy intake (kcal/day)	1671 (1384; 2021)	1708 (1367; 1975)	1634 (1358; 1903)	1737 (1432; 2150)	0.0452
BMI-SDS	−0.39 (−0.97; 0.20)	−0.35 (−0.93; 0.17)	−0.58 (−1.14; 0.12)	−0.24 (−0.93; 0.35)	0.0277
Socioeconomic status					0.0058
low	7 (1.8)	3 (2.1)	4 (3.6)	0 (0.0)	
middle	94 (24.1)	32 (22.1)	37 (33.6)	25 (18.5)	
high	278 (71.3)	105 (72.4)	67 (60.9)	106 (78.5)	
Physical activity (MET-min)	2.9 (1.9; 4.1)	2.7 (1.8; 4.0)	2.9 (1.6; 4.3)	3.0 (2.0; 4.1)	0.7623

Values are n (%) or median (Q1; Q3). ^1^ Kruskal-Wallis test, Fisher’s Exact test or Chi^2^-test. BMI-SDS = standard deviation score of body mass index. MET-min = metabolic equivalent task-minutes.

**Table 3 nutrients-14-04010-t003:** Total food costs and food group costs (EUR/day; EUR/1000 kcal) among participants of the VeChi Youth Study (n = 390), stratified by diet group.

Food Costs		Vegetarian	Vegan	Omnivore	*p*-Value	Pairwise Comparisons		
						VG vs. VN	VG vs. OM	VN vs. OM
Total food costs	EUR/day	4.37 (3.35; 5.45)	4.79 (3.99; 5.94)	4.75 (3.90; 6.42)	0.0036	0.0030	0.0100	0.6521
	EUR/1000 kcal	2.52 (2.12; 3.16)	2.98 (2.51; 3.52)	2.83 (2.30; 3.24)	0.0087	0.0036	0.0391	0.4141
Vegetables ^1^	EUR/day	0.36 (0.24; 0.59)	0.47 (0.29; 0.72)	0.38 (0.19; 0.58)	0.0003	0.0006	0.7177	0.0003
	EUR/1000 kcal	8.52 (6.20; 10.75)	8.23 (6.44; 10.15	8.63 (6.54; 11.44)	0.7835	0.9633	0.5795	0.5795
Fruits ^1^	EUR/day	0.56 (0.29; 0.83)	0.78 (0.51; 1.21)	0.45 (0.28; 0.84)	0.0003	0.0003	0.9674	0.0003
	EUR/1000 kcal	3.53 (2.70; 4.89)	4.03 (3.16; 5.46)	3.53 (2.75; 4.85)	0.7994	0.5467	0.8135	0.7105
Starchy foods ^1^	EUR/day	0.72 (0.51; 0.97)	0.80 (0.52; 1.02)	0.60 (0.42; 0.91)	0.0003	0.1561	0.0026	0.0003
	EUR/1000 kcal	1.19 (0.98; 1.53)	1.26 (0.96; 1.61)	1.13 (0.85; 1.52)	0.4548	0.6270	0.4548	0.2239
Legumes/nuts/seeds ^2^	EUR/day	0.06 (0; 0.20)	0.18 (0.07; 0.46)	0.00 (0.00; 0.10)	0.0003	0.0003	0.0003	0.0003
	EUR/1000 kcal	1.60 (0; 2.50)	1.92 (1.29; 2.41)	0.00 (0.00; 1.82)	0.0003	0.0331	0.0003	0.0003
Dairy ^1^	EUR/day	0.26 (0.02; 0.51)		0.42 (0.27; 0.66)			0.0150	
	EUR/1000 kcal	1.62 (1.23; 2.03)		1.95 (1.55; 2.29)			0.0011	
Meat/fish	EUR/day			0.57 (0.26; 1.11)				
	EUR/1000 kcal			3.82 (3.02; 5.16)				
Eggs ^2^	EUR/day	0.00 (0.00; 0.06)		0.02 (0.00; 0.09)			0.0184	
	EUR/1000 kcal	0.92 (0.0; 2.49)		2.49 (0.0; 2.49)			0.0132	
Dairy alternatives ^2^	EUR/day	0.08 (0.00; 0.35)	0.45 (0.22; 0.69)	0.00 (0.00; 0.00)	0.0003	0.0003	0.0003	0.0003
	EUR/1000 kcal	2.75 (0.00; 4.54)	4.61 (3.31; 5.75)	0.00 (0.00; 0.00)	0.0003	0.0003	0.0003	0.0003
Meat alternatives ^2^	EUR/day	0.17 (0.00; 0.61)	0.25 (0.07; 0.65)	0.00 (0.00; 0.00)	0.0003	0.1142	0.0003	0.0003
	EUR/1000 kcal	5.18 (0.00; 7.80)	5.11 (3.57; 7.57)	0.00 (0.00; 0.00)	0.0003	0.5703	0.0003	0.0003
Convenience foods ^2^	EUR/day	0.05 (0; 0.34)	0.17 (0.02; 0.39)	0.05 (0; 0.4)	0.0865	0.1055	0.9693	0.1623
	EUR/1000 kcal	2.97 (0.00; 5.12)	4.43 (1.47; 6.25)	1.88 (0.00; 4.44)	0.0088	0.0391	0.2911	0.0036
Oils/fats ^1^	EUR/day	0.08 (0.03; 0.13)	0.06 (0.02; 0.13)	0.10 (0.04; 0.15)	0.6521	0.7324	0.5772	0.4056
	EUR/1000 kcal	0.67 (0.49; 0.82)	0.61 (0.42; 0.87)	0.69 (0.53; 0.88)	0.1539	0.5330	0.2084	0.0650
Sweet and snack foods ^1^	EUR/day	0.48 (0.27; 0.89)	0.42 (0.18; 0.77)	0.59 (0.32; 1.02)	0.0781	0.2766	0.2397	0.0261
	EUR/1000 kcal	2.11 (1.68; 2.70)	2.41 (1.85; 3.28	2.41 (1.82; 2.89)	0.0150	0.0042	0.2852	0.0934
Beverages ^1^	EUR/day	0.08 (0.00; 0.87)	0.00 (0.00; 0.64)	0.57 (0.00; 1.51)	0.0045	0.4216	0.0155	0.0022
	EUR/1000 kcal	0.00 (0.00; 7.86)	0.00 (0.00; 8.15)	2.72(0.00; 23.67)	0.1460	0.9317	0.0934	0.1055
Other ^2^	EUR/day	0.01 (0.00; 0.02)	0.01 (0.01; 0.03)	0.01 (0.00; 0.01)	0.0003	0.0011	0.6518	0.0003
	EUR/1000 kcal	2.45 (1.41; 4.46)	2.86 (1.74; 4.90)	2.16 (1.49; 3.98)	0.1309	0.2719	0.5028	0.0404

VG: Vegetarian; VN: vegan, OM: omnivore. *p*-values corrected using false discovery rate (Benjamini–Hochberg method). ^1^ Ancova. ^2^ Kruskal-Wallis test and Wilcoxon-Mann-Whitney test for pairwise comparisons.

## Data Availability

The data described in the manuscript, code book and analytic code will not be made available because of data protection regulations.

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
