# Peer review of "Food Costs of Children and Adolescents Consuming Vegetarian, Vegan or Omnivore Diets: Results of the Cross-Sectional VeChi Youth Study"

_nutrients, 2022, doi:10.3390/nu14194010_

Round 1
Reviewer 1 Report
In 2.4 food price collection section, the paper should add the representativeness and reasonableness of the data. Due to the covid pandemic restrictions, it was hard to collect the prices in retail stores. Thus, the paper collected food price mainly from the websites. However, it is not known whether the food prices collected from the website are representative.
It would be clearer to add “in the same point of sale” to line121. The sentence can be modified as “If more than one kind of similar foods in the same point of sale was offered …”
In 3.1 sample characteristics section (line 193), the paper mentioned “The SES differed significantly between the diet groups (Fisher’s Exact Test, p=0.0155).” The contradiction was table2 show p=0.0058.
Author Response
We thank the reviewer for her/his careful review of the manuscript and followed all suggestions:
In 2.4 food price collection section, the paper should add the representativeness and reasonableness of the data. Due to the covid pandemic restrictions, it was hard to collect the prices in retail stores. Thus, the paper collected food price mainly from the websites. However, it is not known whether the food prices collected from the website are representative.
Thank you for your comment. To more explain the reasonableness of our approach, we added the following sentence:
“With this approach, we captured foods that have an impact on costs due to their high quantity eaten, as well as foods that are eaten only in small quantities.”
It would be clearer to add “in the same point of sale” to line121. The sentence can be modified as “If more than one kind of similar foods in the same point of sale was offered …”
Thank you for this valuable comment. We changed the sentence accordingly.
In 3.1 sample characteristics section (line 193), the paper mentioned “The SES differed significantly between the diet groups (Fisher’s Exact Test, p=0.0155).” The contradiction was table2 show p=0.0058.
We are sorry for this typo (reflecting the p-value before adjusting for multiple testing) and changed the p-value in the text accordingly.
Author Response
Reviewer 2
We thank the reviewer for her/his careful review of the manuscript and apologize for the grammatical errors she/he found
This was an interesting and original study.
Thank you for this comment.
My comments are minor. Minor grammar issues including
line 45 – pattern(s): Changed as suggested
line 46 – plant-based diet/s and omnivorous diet/s should be consistently singular or plural. Plural seems preferable to reflect the variety of diets under each of these categories: Changed as suggested
Line 52 – pattern(s) : Changed as suggested
Lines 60-61 – as outdated seems unnecessary the second time it is used in this sentence: Changed as suggested
Lines 222-223 – with (the) exception of: Changed as suggested
Line 225 – statistical(ly) significant: Changed as suggested
Line 235 – replace more with longer: Changed as suggested
Line 236 – vegetables were: Changed as suggested
Line 249 – did not differ significantly: Changed as suggested
Line 310 – replace to mention with mentioning: Changed as suggested
Line 313-314 – reword to improve clarity: Changed as follows: “for example by selection of less expensive foods within a food group”
Line 313 – partial(ly): Changed as suggested
Line 343 – to provide sufficient protein, iron, and zinc: Changed as suggested
References need to use a consistent format: References were checked
Lines 67-68 – It’s unclear what the meaning of this sentence is. Is this referring to the replacement of animal products with plant-based products among omnivores? Perhaps rewording would help to improve clarity: As suggested, we extended the sentence: “and replaced by plant foods”.
What is “germ” in the last line of Table 1? Wheat germ? We have deleted this term.
Lines 249-251 – please reword sentence to make its meaning clearer: Reworded as follows: “Food costs for convenience foods did not differ significantly. When standardised for energy intake from the respective food group the difference between vegans (4.43 €/1000 kcal) and both other groups (vegetarian: 2.97 €/1000 kcal; p=0.0391 and omnivore: 1.88 €/1000 kcal; p=0.0036) was significant”
Line 278 – Consider replacing “had to spend” with spent. The families were not required to spend this amount of money.: Changed as suggested
Lines 366-367 – clarify what is meant by “animal source food assuring special animal welfare.” Perhaps provide an example. Are you referring to, for example, cage-free or free range products? Sentence is expanded by the following words: “animal source food assuring special animal welfare, e.g. pasture milk or meat from animals raised in larger stables,...”